# T.S. Eliot in the 1918 Pandemic: Abjection and Immunity

## Huiming Liu

School of Literatures, Languages and Cultures, University of Edinburgh, Edinburgh EH8 9YL, UK;
s1824150@ed.ac.uk

**Abstract:** The influence of the 1918 pandemic was overshadowed by the catastrophe of the First World War. The current COVID-19 pandemic leads the academic attention to how the 1918 pandemic shaped literature of that period. Elizabeth Outka's book brings the history of the pandemic into the study of modernism. The vast scale of a sudden outbreak of pandemic disease had made decent burials and mourning very difficult. Outka argues that *The Waste Land* mourns the deaths during the pandemic. The traumatic experience of the pandemic can also be found in the difficulty of speech and the fragmentation of ghostly existence in *The Waste Land*. Building upon Outka's work, this essay will engage with the cultural influences of the pandemic in Eliot's other works and reveal how the famous touchstones of modernisms are shaped by such an event. I will specify how the war and the pandemic were connected in the following section on historical backgrounds. Immunity aims to fight against foreign invaders such as viruses on a micro-level. However, on a macro-level of politics, the logic of the immune system often wrongly identifies certain groups as the scapegoats for contagious diseases. My article aims to reveal the underlying metaphor of immunity in Eliot's writing of the abject in the late 1910s. By doing so, I hope to contribute to current academic discussions of Eliot and the writing of the pandemic, anti-Semitism and post-colonialism.

**Keywords:** pandemic; T.S. Eliot; modernism; gender; race; WWI

The influence of the 1918 pandemic was overshadowed by the catastrophe of the First World War. The current pandemic of COVID-19 contributes to more academic discussion on how the 1918 pandemic shaped literature of that period. Elizabeth Outka brings the history of the pandemic into the study of modernism (Outka 2018, pp. 142–56). The vast scale of a sudden outbreak of pandemic disease had made decent burials and mourning very difficult. Outka argues that *The Waste Land* mourns deaths during the pandemic. The traumatic experience of the pandemic can also be found in the difficulty of speech and the fragmentation of ghostly existence in *The Waste Land*. Building upon Outka's work, this essay will engage with the cultural influences of the pandemic in Eliot's other works and reveal how the famous touchstones of modernism are shaped by such an event.

## 1. Immunity and Abjection

Drawing on Maebh Long's recent research into immunity and abjection, my analysis of Eliot's writing focuses on the affects related to the pandemic. Long's research begins by clarifying the etymology of the word "immunity", which means exemption from public service (Long 2020, pp. 220–25). Later, she points out that the discovery of the phagocytes (cells that protect the host by devouring and destroying antigens and inoculations) by Elias Metchnikoff in the late nineteenth century was an advance in immunology that influenced the social and cultural understanding of a self-defensive body. Paul Erhlich (1854–1915) proposed the dictum *horror autotoxicus*, which argues for the possibility of the body attacking itself due to the immune system's mis-identification of the healthy body tissues as invading pathogens (Silverstein 2001, p. 1). The difference between the contemporary understanding of autoimmunity and Erhlich's *horrorautoxicius* is that Erhlich did not see the possibility in self-destruction during the formation of antibodies against

oneself. It seems that Erhlich believed that the auto-immune system of the body would eventually heal itself.

I will specify how the war and the pandemic were connected in the following section on historical backgrounds. Immunity aims to fight against foreign invaders such as viruses on a micro-level. However, on a macro-level of politics, the logic of the immune system often wrongly identifies certain groups as the scapegoats for contagious diseases. My article aims to reveal the underlying metaphor of immunity in Eliot's writing of the abject in the late 1910s. By doing so, I hope to contribute to current academic discussions of Eliot and the writing of the pandemic, anti-Semitism and post-colonialism.

## 2. Historical Background

The symptoms of the 1918 pandemic (also called the "Spanish flu") varied among people. Light symptoms of the pandemic included coughing, loss of taste or smell, fever, pain, fatigue and sleepiness. In more serious cases, coughing developed into breathing difficulties, which could result in cyanosis (bluish skin color) due to the lack of oxygen in one's body. The abovementioned symptoms can be found in John Barry's and Laura Spinney's works on the history of the pandemic (Barry 2005, p. 236; Spinney 2017, p. 45). There were cases of depression and even suicides after delirium (ibid., p. 238).Some suffered from sleepiness and low energy after they recovered (ibid., p. 232). Vivien Eliot describes T.S. Eliot's symptoms of fever and long-lasting sleepiness during the influenza in her letter on December 15th in 1918 (Eliot et al. 2009, vol. 1, p. 6322). Based on the timing and the symptoms it is likely that Eliot contracted the disease during the pandemic. As for the cause of the pandemic, there were various explanations. At first it was thought to be merely a small flu. It was misdiagnosed as typhoid, typhus (also known as trench fever transmitted by lice), cholera, or even the bubonic plague (Barry 2005, pp. 235, 266, 376). During the 1910s, doctors did not have the means to know that the cause was the H1N1 virus. Most scientists thought that the cause was a bacterium (Van Epps 2006, p. 803). At that time, a virus was understood as a form of toxin smaller than bacteria: "Whether virus is alive or not is kind of debatable. It is either a very complex chemical or a very simple life form" (Arnold 2018, p. 209). Viruses stood on the biological/chemical, organic/inorganic and alive/death boundary.

The virus took various forms in the social and literary imagination, such as miasma and swamp gas (Barry 2005, p. 50). Miasma was regarded as the wind of illness. Swamp gas was related to filth and excrement. There was also a belief that the ill winds were "war-bred pestilence" (ibid., p. 735). The connection between the pandemic and the Great War likely came from the first outbreaks in the military camps in several countries, including the United States, France and the United Kingdom. The outbreaks were particularly linked with transportation by sea because international shipping facilitated the spread of the disease. Another explanation for the origin of the disease was that it came from the animals on farms because there were outbreaks reported in village farms in the U.S. (Arnold 2018, p. 31). The pandemic created an atmosphere of paranoia and fear concerning the unidentified cause.

Such paranoia and fear about the disease sometimes turned into scapegoating of certain groups. For example, sex workers were one of the targeted groups for the spread of the pandemic. Apart from the violation of quarantine measures, they were blamed for the transmission of venereal disease during the Great War among soldiers. H.G. Irving warned of the threat to public health of venereal diseases (Irvine 1918, pp. 1029–36). The misogynistic discourse of the disease can be seen in calling the pandemic the "Spanish Lady":

> As the first wave of Spanish flu broke across Europe in June 1918, cartoons and illustrations appeared depicting the flu as the "Spanish Lady". Spanish flu was personified as a death-headed, skeletal woman in a black flamenco dress, complete with mantilla and fan (Figure 1; Wilhelm 1918). Such a depiction implied that the Spanish lady was a prostitute, free with her favors and infecting everybody at the same time (Arnold 2018, p. 342).

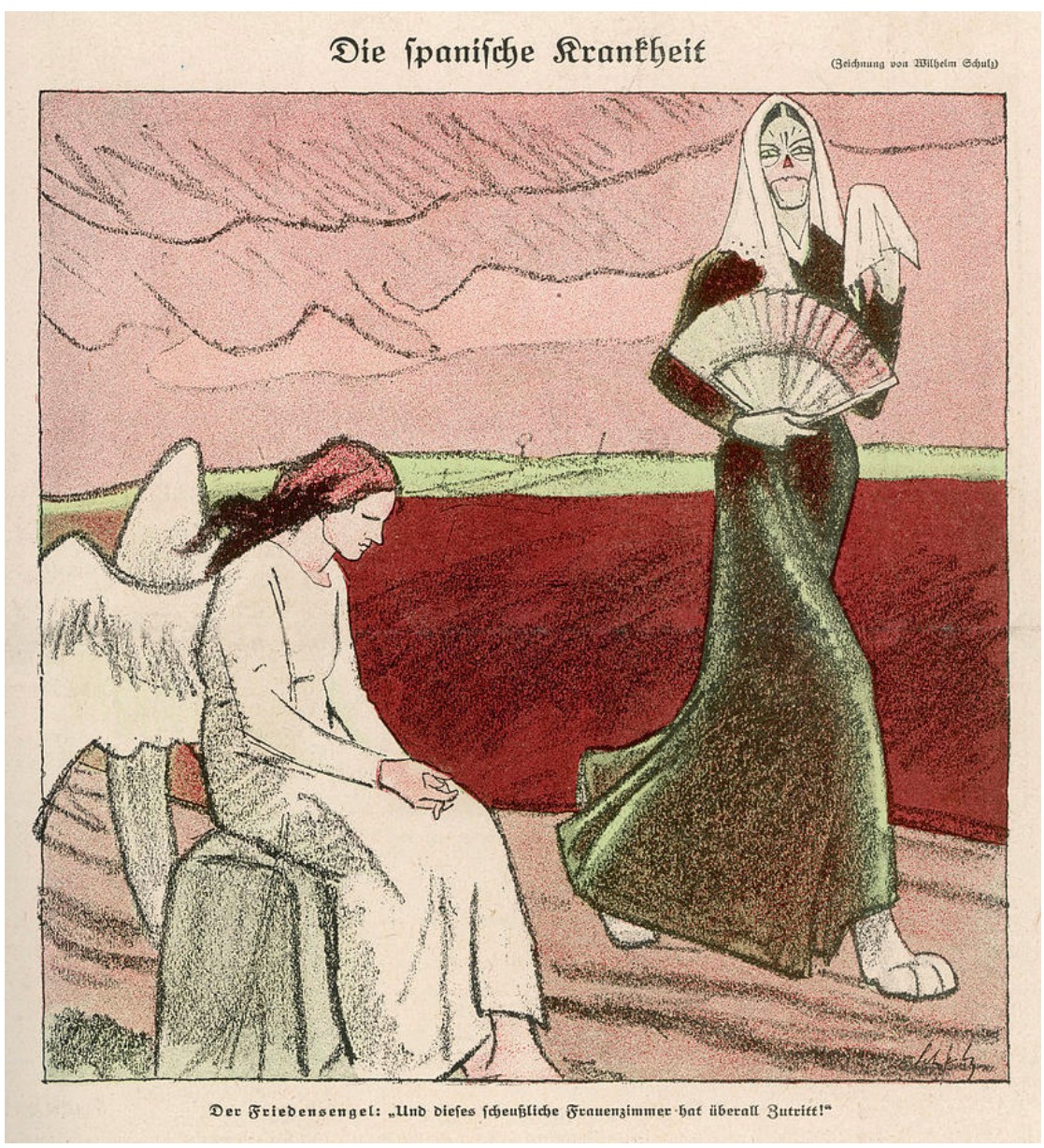

**Figure 1.** Schulz Wilhelm. The "Spanish Flu" Epidemic Overtakes the Angel of Peace. 1918.

Richard Collier's book on the 1918 pandemic was titled *Plague of the Spanish Lady*. The book's cover is the picture shown above. The "Spanish lady" in black overtakes the angel of peace. Spanish flu was personified as a dangerous and bestial woman with her foot portrayed asa paw. "Spanish Lady" is a misogynistic representation of the pandemic in a male-centered society that projected its fear and paranoia against the pandemic onto women. The picture above captures the misogynistic imagination of the flu in a patriarchal society. Immunity was therefore defined by a male-centered society that wrongly identified women's sexuality as the cause of the abject state in the pandemic. The scientific sense of immunity may sound indifferent and detached, but the social significance of immunity can be defined by the dominant ideologies.

An imperial immune system also tried to identify the cause of disease from the perspective of the war and its colonies. The stigmatization of soldiers can be linked to the hospitals in navy bases where the outbreaks were found in the U.S. and Europe. In Étaplethere was an outbreak of the influenza at the navy hospital (Arnold 2018, p. 18). British troops in France also suffered from the mysterious illness (ibid., p. 53). There was even a conspiracy theory that regarded the German U-boats as the disseminators of the

disease (ibid., p. 14). The link between disease and sea travel not only revealed the terror of the war, but also contributed to doubts surrounding trade and the empire. According to the First Sea Lord Admiral, the influenza was spreading in the British Navy in January 1918 and as a result the soldiers could not protect the merchant ships on the sea (ibid., p. 58). Britain terminated trades with its colonies at that time. The pandemic outbreaks in India shattered the imperialist view of a tropical paradise of "banana trees and coconut palms" (ibid., p. 91). The "tropical paradise" turned into an abyss of disease. As postcolonial thinkers from Said to Fanon have observed, the colonies were viewed by the empire as either a promising land of resources or a backward and chaotic place for "advanced" colonial regulations. Said comments directly in *Orientalism*: "Orientals were always and only the human material one governed in British colonies" (Said 1978, p. 48). In the colonizers' eyes, due to the abject pandemic, the colonies turned into a threatening space whose resources were out of their control. The fear of being devoured by the chaos and disease of the colonies was embedded in the imperialistic system. H.G. Wells' novel *The War of the Worlds* (1897) was an example of the imperialistic anxiety. In the novel, the Martians invade Britain but at the end the invaders are killed by a new disease to which they are not immune. Wells had in mind the genocide of the Tasmanian people in contact with a new disease brought by the British invaders (Wells 2017, p. 162). The failure of the Martians mirrored the collapsing British Empire in the face of a new disease. The plot of the novel revealed the imperialists' anxiety over managing new disease within the rule of the empire. The pandemic contributed to the anti-colonial movements despite its submerged presence in the historical narratives.

Immigrants and minority groups were also the targeted scapegoats for various diseases. In April 1918, Germany closed its borders to Jewish immigrants from Eastern Europe until the end of the war, citing the danger of epidemics such as typhus (Bergmann and Ulrich 2017, p. 8). Whether it was typhus or the "Spanish flu" was of no significance because disease had become a slippery signifier of fear against certain groups of people. Even the scientists only knew that the cause of the pandemic was a kind of filterable poison smaller than bacteria. Given the chaos in understanding the pandemic at that time as discussed above, the late 1910s was a gloomy period when the advancement of science and technology could no longer explain the massive scale of death by disease, war or both. Society tried to understand and explain the disease in different ways and during this process they invoked prejudices out of fear and paranoia. Modernist literature was written in the same period of various unknown diseases. The social and scientific discourses were present in modernism. At the same time authors such asEliot also tried to represent disease, abjection and immunity in their works. Having established the historical background and the cultural discourses on disease and immunity, I will proceed to discuss Eliot's writing from these perspectives.

### 3. "Sweeney among the Nightingales": Empire and the Pandemic

My analysis of "Sweeney Among the Nightingales" focuses on the abject, immunity and empire in the context of the pandemic. The poem was composed in mid-1918 and published in September of the same year. The 1918 pandemic broke out in June. "Sweeney among the Nightingales" was very likely composed under the impact of the "Spanish flu". The poem is set up with a gloomy atmosphere of paranoia: "Death and the raven drift above"; "Gloomy Orion and the Dog/Are veiled" (l. 7, 9–10). The image of a veil corresponds to the use of facemasks during the pandemic. Despite people's unease with blurred identities in public spaces, the gravity of the pandemic eventually made masks a necessary choice. Eliot refers to the pandemic in a letter on July 7th in 1918: "We have been living on quietly and trying to escape the 'Spanish influenza' so called" (Eliot et al. 2009, vol. 1, p. 5629). The poetic speaker deliberately points out the Spanish identity of a woman "in the Spanish cape" (l. 11). The poem focuses on her feminine sexuality by gazing at her seductive gestures: "The person in the Spanish cape/tries to sit on Sweeney's knees"; "She yawns and draws a stocking up (l. 11–12, 16). The character Sweeney appears multiple times in Eliot's works in brothels. Eliot always associates

Sweeney with feminine sexuality. The depiction of feminine sexuality here is similar to the male-centered paranoiac depiction of the "Spanish lady" in that she is described to be in league with another woman with "murderous paws": "Rachel *née* Rabinovitch/Tears at the grapes with murderous paws/She and the lady in the cape/Are suspect, thought to be in league" (ll. 23–26). The poem centers on the two women's threatening sexuality and their plotted murder in the bloody wood: "And sang within the bloody wood/When Agamemnon cried aloud" (ll. 37–38). Agamemnon is the Greek soldier who was murdered by his wife Clytemnestra. The reference to him in the poem suggests a similar plot of bloody murder by the women. "Hence the poem seems to reveal male paranoia against women's sexuality which was appropriated for projecting fear of the infectious "Spanish flu" in a male-centered society. Women are bestialized as threatening animals with dangerous paws. The similarity between the poem and the "Spanish lady" shows the misogynistic representation of women during the pandemic. It is not certain whether Eliot had seen the image of the "Spanish lady" by Wilhelm Schulz. Sacre Cœur in Paris also is in Montmartre, often associated with brothels. However, according to Catherine Arnold, the images of the Spanish Lady and face masks were widely circulated in publications in Europe during the pandemic (Arnold 2018, p. 342). Hence, it is likely that Eliot has been influenced by such misogynistic depiction.

It is also worth pointing out that the prostitutes Are viewed as a threat to the military character Sweeney. The zebra stripes swelling along Sweeney's jaw suggest a tight and restrictive body guarded against the environment. Vincent Sherry suggests that Sweeney is a soldier in a tight uniform that swells on his neck: "the zebra stripes along his jaw, swelling to maculate giraffe"; "and Sweeney guards the horned gate" (ll.3–4, 8) (Sherry 2015, p. 257). Yet, zebra and giraffe here seem to suggest that Sweeney's clothes swell upward to the face, which could be seen as an attempt to mask himself. Sweeney's identity as a soldier with prostitutes also reflects the public stigmatization of venereal disease for the loss of morale during the war. Women's sexuality becomes the abject scapegoat for diseases in general during the war and the pandemic. Women's sexuality is viewed as the abject that challenges the male body's health. Sweeney's character as a soldier also links the healthy male body to military empire in this poem.

The poem invites readers to think about empire and colonialism through various details. First, the poem directly mentions the River Plate, which is located in both Argentina and Uruguay (former colonies of the Spanish empire): "The circles of the stormy moon, slide westward toward the River Plate" (ll. 5–6). The person in a Spanish cape also points to the context of the Spanish-speaking countries alongside the River Plate. The poem also mentions Catholicism: "The nightingales are singing near, the Convent of the Sacred Heart" (l. 35–36). Catholicism in America is also a result of the Spanish colonization. The poem not only refers to the Spanish empire, but also the British empire. "The River Plate" derives from the Spanish name Rio de la Plata (the river of silver) assigned by the colonizers. "The River Plate" is the British empire's colonial name for the area. In 1806, the British empire attempted to colonize the region but eventually failed. Sweeney as a soldier also likely comes from Anglophone countries. Other references to colonialism in the poem include the following single line concerning tropical fruits: "Bananas, figs and hothouse grapes" (ll. 20). According to a bulletin by the Royal Botanic Gardens in 1888, the fruits mentioned in the poem were all classified as "colonial fruits". All these details invite readers to consider the poem from a postcolonial perspective (Royal Botanic Gardens 1888, pp. 1–23). The multiple elements of the Spanish and British empire suggest the potential in analyzing Eliot's writing of the disease through a postcolonial perspective.

The poem is written with a focus on the two women's conspiracy (one in a Spanish cape and the other whose name is Rachel née Rabinovitch). The latter's Jewish name gives an anti-Semitic message. Such a message is clearer in the dehumanized portrayal in the following: "Rachel née Rabinovitch, tears at the grapes with murderous paws" (l. 23–24). Similar to the bestial and misogynistic depiction of the "Spanish lady", the Jewish origin here points to the stereotype of Jewish conspiracy. Anthony Julius also points out Eliot's as-

sumption of an "International Jewish conspiracy" in this poem (Julius 1995, pp. 83–85). The poem associates the murderous conspiracy against Sweeney with the following identities: Jewish, Spanish and feminine. It seems that the poem falls into line with the misogynistic and racist discourse that describes the pandemic as spread by the femme fatale and Jewish people. The "murderous paws" of Rachel née Rabinovitch are an image of the abject that threatens the corporeal existence of Sweeney. The abject during the pandemic was associated with women's sexuality, minority groups or immigrants (in this case Jewish people). Meanwhile the poem also depicts a character that successfully defends himself against the abject.

The poem describes a vertebrate who successfully guards himself after he perceives danger from the women: "The silent vertebrate in brown, contracts and concentrates, withdraws"; "She and the lady in the cape/Are suspect, thought to be in league/Therefore the man with heavy eyes, declines the gambit, shows fatigue" (ll. 21–22, 25–28). The withdrawing silent vertebrate seems to be the same man who declines the gambit. Whether the vertebrate is the same man as Sweeney is not explicitly told in the poem. The vertebrate protects himself from the danger through the tense movement of muscle contraction and concentration. The language in this stanza becomes very embodied in describing physical strength. The word "vertebrate" reminds the readers of the hard strength of bones. The contraction and concentration of muscles is also an image of physical strength in self-guarding. Physical strength suggests a healthy functioning of the immune system. The bone structure and muscle strength also depict human bodies' hard physical form. In this poem, Eliot celebrates the physical strength, the hard frame and the immunity of vertebrates. Such a hard frame is positioned to be under the constant threats of trespassing disease from women's sexuality or Jewish conspiracy. The vertebrate's successful defense against the abject is an ideal view of a healthy immune system which can maintain the self's boundary of the bones and muscles and meanwhile successfully identify and target the other.

Compared with the vertebrate in the poem, Sweeney cannot be as successfully separated from the abject. We are not told clearly in the poem whether he escapes from the murder, but the poem describes the planned criminal scene against him. The trespassing of Sweeney's physical identity is mapped onto the following site of murder: "The host with someone indistinct/Converses at the door apart" (ll. 33–34). The open door, the unidentified guest and the murder can all be seen as the projection of the abject in which state the inside/outside melts down and one's corporeal identity also breaks down. The conspiracy against Sweeney is compared with the murder of Agamemnon at the end of the poem:

The nightingales are singing near

The Convent of the Sacred Heart

And sang within the bloody wood

When Agamemnon cried aloud (ll. 35–38).

When Agamemnon sailed back from the siege of Troy, he was murdered by his wife Clytemnestra. The murder plot against Sweeney is paired with the fall of Agamemnon. Eliot provides a framework from ancient Greek mythology for the story of the poem. Agamemnon as a soldier sails to Troy and wins the battle. However, he is murdered by his wife upon his arrival back home. Sweeney as a soldier sails to the River Plate. He is also threatened with death by women's sexuality. The poem attempts to frame the colonial experiences into the ancient Greek myth of Agamemnon. Such an attempt can not only be seen in the mapping of the story but also the description of celestial signs in the first few stanzas:

The circles of the stormy moon

Slide westward toward the River Plate,

Death and the Raven drift above

And Sweeney guards the horned gate.

Gloomy Orion and the Dog

Are veiled; and hushed the shrunken seas (ll. 5–10).

In ancient Greek mythology, celestial signs are major contextual signs describing characters' fates. For example, Agamemnon mentions the star Pleiades in relation to the stormy weather and refers to the gloomy war (Pfundstein 2003, p. 400). Eliot also inherits such a celestial contextualization in his poem, but the astronomical signification varies from the ancient Greek.

The astronomical description in the poem can be related to disease. In the nineteenth century, the British navy in their colonies believed that the moon could have an influence on people's health (Harrison 2000, p. 25). Mark Harrison discusses how the sun and the moon were associated with disease in the eighteenth and nineteenth centuries. Harrison traces the astrological explanation of diseases to Newton's theory of gravitation which established an analogy between "the action of the Sun and Moon over the waters of the Earth and their presumed action upon the fluids of the human body" (ibid., p. 29). Lacking empirical evidence, sometimes the conclusions were opposite to each other in terms of whether the waxing or waning of the moon increased chances of fever. What is certain was that human bodies were imagined in terms of atmosphere or air, which was influenced by the sol-lunar cycles. In the early nineteenth century, the theory thrived among British army physicians who believed that tropical diseases such as fever were likely to occur during periods of a full or new moon (ibid., pp. 35–37). In the poem, the shrunken seas are described as being hushed under the full moon. The description fits into the theory of the sol-lunar impacts on the atmosphere causing the seas to rise or shrink. The stormy cycles were often associated with fever and other diseases. The Dog Star was also believed to bring cases of fever before the late nineteenth century (ibid., p. 31). The veiled Dog Star in the poem also corresponds to the mask-wearing activities on the ground. The poem's description of the shrunken seas under the stormy moon corresponds to the sol-lunar influences on the atmosphere above the sea. Atmosphere and disease are closely linked together in the theory of miasma. During the 1918 pandemic, miasma theory was still popular among people who believed that winds or air caused illness. Considering the time when the poem was written, the relation between the astrological description and diseases seems even stronger.

Hence, the poem writes disease and immunity from past and present discourses, including the scientific description of vertebrate and muscle tension, the social and cultural portrayal of the "Spanish flu" and the folk beliefs of astrological signification. However, all these aspects are based on an immune and healthy empire. The poetic speaker targets the "Spanish lady" as the object to be immunized. Although the silent vertebrate successfully immunizes himself from the murder, the poem depicts a state of abjection in which the empire fails to distinguish itself from the colonies. Unlike the clear and hard physical boundary of the vertebrate, the empire's boundary gradually melts into an abject state in which the empire's soldier Sweeney can be murdered because of feminine temptation. The poem reveals the fear of femininity which underlies the ideology of the empire. Kristeva discusses the relationship between the motif of feminine temptation and the abject:

> The other place sit (feminine temptation) within the femininity-desire-food-abjection series. The story of the fall sets up a diabolical otherness in relation to the divine . . . Adam is no longer endowed with the composed nature of paradisiac man, he is torn by covetous desire: desire for woman—sexual covetousness since the serpent is its master, consuming desire for food since the apple is its object. He must protect himself from that sinful food that consumes him and that he craves (Kristeva 1982, pp. 127–216).

"Sweeney" is written within the femininity-desire-food-abjection series. In the previous paragraphs I have established that the conspiracy of the "Spanish lady" reveals the fear of feminine temptation to overturn the male-centered self-identity. Feminine sexuality in the poem is described along with the following food and fruits: coffee, bananas, figs

and hothouse grapes. The poetic speaker and the silent vertebrate detect the conspiracy when Rachel nee Rabinovitch's "murderous paws" tear at the grapes. The scene is similar to the story of the fall. The desire for both women and food will consume the masculine figures (Adam and Sweeney) and break down their male-centered identity. Another scene of conspiracy involves the Spanish lady and coffee. She deliberately overturns a coffee-cup and provocatively draws her stocking up: "Slips and pulls the table cloth/Overturns a coffee-cup/Reorganized upon the floor/She yawns and draws a stocking up" (l. 9–12). Sexuality and food are regarded as potential threats to masculine identity in the poem because they suggest the consuming abjection that can murder the men in the form of disease. Such a depiction of the Spanish lady matches the feminization of fatal disease during the pandemic. The poetic speaker projects the anxiety and fear of "feminine abjection" onto the tropical food. The fear of "feminine abjection" can thus be seen as the fear of losing control over the empire. Tropical fruits and coffee are resources from the colonies of both the British and Spanish empire. The woman tearing at the grapes alerts Sweeney because she violently occupies and consumes the resources. The woman covetously tearing the grapes can be paralleled with Eve eating the fruits in the story of the fall. Her action reminds the male persona of murder and bestiality. The paranoia against femininity in the poem is similar to the scapegoating of Eve for the fall of men. The fruits in the story of the fall are not only linked with feminine desires but also connected with colonialism as shown before. Hence, the fear of feminine temptation is also mapped onto colonies. The male poetic persona and Sweeney are afraid of being consumed by the desires for colonial resources.

The consumption of the tropical fruits is linked to the mortality of Sweeney in the poem, which seems to reproduce the colonial dietary anxieties: "One of the most important reasons diet was so closely bound up with colonial anxiety was that it appeared to explain the extraordinary mortality and morbidity rates of Europeans in tropical climates" (Bewell 2000, p. 150). There were returning soldiers from tropical areas who were often seen as the survivors of tropical diseases with a weakened constitution (Bewell 2000, p. 278). The poem's association between colonial fruits and mortality can be seen in the colonizers' fear of being consumed by the colonies. In order to defend the masculinity of empire, the poem establishes an underlying immune system which targets women, Jewish people and colonies as the pathogenic "other". However, the ending of the poem seems to reveal the state of abjection in which the murder is committed and the boundaries between empire and colonies are broken.

The ending of the poem writes the abject with the direct images of blood and excrement:

The nightingales are singing near

The Convent of the Sacred Heart,

And sang within the bloody wood

When Agamemnon cried aloud

And let their liquid siftings fall

To stain the stiff dishonoured shroud (ll. 35–40).

Blood and excrement are classified as the abject because they directly challenge one's subjective identity as an individual living being. Eliot uses a subtle image to describe the "bloody wood" in the previous stanza: "Leaning in/Branches of wisteria/Circumscribe a golden grin" (ll. 30–32). The bloody wood of wisteria invades the "indoor space" and commits murder with a "golden grin" from the ominous moon. Here, the abject is described as a branch of intruding wisteria which disrupts one's identity as a living being. It seems that Eliot does not want to cleanse the abject in catharsis as Kristeva argues about literature's engagement with abjection. Instead, he immerses himself in the abject and attempts to absorb it into the harmonizing form of his poem and his ideal of tradition.

Kristeva elaborates on the relationship between abjection and catharsis throughout the western philosophical tradition. The Platonic catharsis, according to Kristeva, aims at debasing the impurities of abjection and escaping into the ideal world. Aristotelian catharsis, on the other hand, looks for the harmony between poetic forms (such as rhythms and

songs) and abjection (in this case sexuality). Kristeva thinks that these were the two models of thinking about catharsis and abjection until Freud's psycho-analytic approach emerged:

> The completely mimetic identification (transference and counter-transference) of the analyst with respect to analysands . . . allows one to regress back to the affects that can be heard in the breaks in discourse, to provide rhythm, too, to concatenate (is that what "to become conscious" means?) the gaps of a speech saddened because it turned its back on its abject meaning (Kristeva 1982, pp. 30–31).

Kristeva sees the possibility to speak artistically about the abject through psychoanalytic dialogue between two people's attempt to mutually identify with each other. However, in Eliot's "Sweeney" poem, the poet's writing of the abject falls into the more classical Aristotelian module which attempts to forcibly "harmonize" the abject into a poetic form. Such a form can be seen in the image of the nightingales.

Nightingales are a classic literary symbol of rhythmic sound. Eliot acknowledges that he does not know in reality how the nightingales sound, but only in their literary association (Eliot et al. 2015, vol. 1, p. 545). Nightingales' sound appears in many literary works as a cathartic symbol. For example, the story of Agamemnon's murder by his wife is accompanied by the sound of nightingales. As mentioned before, Agamemnon's death is an over-arching structure of the story in this poem. Eliot directly mentions the nightingales' singing at the site of Agamemnon's death in the poem. He also points out the reality that nightingales should not be present in the season where Agamemnon was killed (ibid.). Hence, the parallel use of nightingales' singing in this poem can be regarded as an attempt to cathartically harmonize with the final murderous scene. The sound pattern (meter) of the poem is also fairly stable with iambic tetrameter and a ballad rhyme scheme of abcb. Hence, Eliot attempts to absorb the abject disease, sexuality and colonies into the traditional form of Greek mythology and a cathartic harmonizing poetic meter. Such an attempt reveals the desire to manage the male/Euro-centric desire for feminine sexuality and colonial resources by stigmatizing them as the threats and filth which can carry and spread diseases. However, the nightingales' sound in the poem fails to convey catharsis as in the Greek mythology. Instead, the scatological description ("their liquid siftings fall/To stain the stiff dishonoured shroud") of the nightingales' feces reveals the disharmony and abjection between empires and colonies. The cathartic symbol of nightingales was also imaginatively used by the Europeans to "harmonize" with their foreign colonies. Christopher Columbus mistook the sound of native birds in the American continent for the sound of nightingales: "And the nightingale was singing and other birds of a thousand kinds in the month of November there where I went" (Levine 2016, vol. 1. p. 60). In reality, there should not be any nightingale singing at that time in the American continent. Columbus's automatic mapping of the nightingales' sound onto the foreign land can be seen as his self-defense mechanism that attempts to orient himself in the colonies. Nightingales' sound becomes a harmonizing bridge between Europe and the colony and thus it cleanses Columbus's fear of being cast off (abject) into the foreign land. His imagination reveals a Euro-centric ambition to absorb everything foreign into his cultural tradition.

Eliot was likely familiar with the story of Columbus, given that he wrote a "Colombo" poem to ridicule Columbus in relation to issues of sex, race and colonialism. Sweeney is also ridiculed in the end with the nightingales dropping their "liquid siftings" to "stain the stiff dishonoured shroud" (ll. 39–40). The scatological reference immediately breaks the nightingales' literary association with catharsis. According to Barry (2005), from the Victorian age to the early twentieth century, "swamp gas" was the euphemism for excrement, which was regarded as the cause of various infectious diseases. Bird excrement is linked with the abject because it was not only regarded as part of the "swamp gas" that could cause disease but also challenges the very foundation of imperialism. The bird excrement mentioned in the poem can be related to the British colonizing trade of guano. The nightingale is also connected to the British archipelago, the Nightingale Islands, where guano was a resource for the British empire. In the poem, Sweeney's shroud of a soldier is stained by the excrement of nightingales. The empire's soldier's honor is broken faced with

the abject image. Hence, the scatological reference in the poem reveals the disharmony between the cathartic form and the abject. Unlike in Columbus's letter, the nightingales in "Sweeney" only emphasize the abjection of the empire's soldier rather than offering any cathartic effects. Although Eliot reveals the colonial anxieties of abjection in the poem, he does not offer an opportunity for a mutual conversation between the poetic speaker and the abject. Instead, the poetic speaker seems to be always battling with the abject in his immune system. The blurred boundary between self and other in one's immune system existed in the concept of autoimmunity in the 1910s. Eliot's poem also reveals the fear of self-consuming in the state of abjection (blurred self/other distinction). Eliot's poetics focuses on an immune form of European literary tradition that seeks to battle with the "pathogens" and absorb them. The "pathogens" are defined as the feminine, Jewish people and empire's colonies.

Gabrielle McIntyre offers an insightful analysis of race, sex and history in Eliot's Bolo and Colombo poems:

> For the most part, the Cuban natives also share the same propensity to anal and excessive eroticism as the colonists, and Columbo's crew thus find uncannily queer mirrors of themselves in the colonies, offering an inverted version of Bhabhan mimicry (McIntire 2002, p. 293).

McIntire's understanding of mimicry can also be found in the intersection between feminine sexuality and colonial food in "Sweeney". However, Eliot's critique of colonialism is relatively weak in "Sweeney" compared with the Bolo and Columbo poems. McIntire's argument centers on the comical affects and the ironic portrayal of the colonizers' excessive physicality in history: "(The poem proposes that) colonial discovery was about sexual conquest, and that to remember the origins of colonialism we would do well to laugh at its physical excesses" (McIntire 2002, p. 286). The Bolo and Colombo poems criticize colonialism through comedy. However, in "Sweeney", the uncanniness of femininity and colonialism develops into life-threatening abjection. Given the history of the pandemic, Eliot conveys the paranoia and horror of the abject situation and expresses the patriarchal and Euro-centric anxieties over femininity, colonialism and "Jewishness". The established intersection between femininity and abjection is also present in "Gerontion" but it is more related to anti-Semitism rather than colonialism.

In "Gerontion", Eliot gives up on the ballad form and depicts the state of abjection and anti-Semitism without a stable poetic form.

## 4. "Gerontion": Anti-Semitism, Money and the Pandemic

"Gerontion" focuses on the viral disease, money and anti-Semitism in its writing of abjection. The presence of disease can be found in the following details of symptoms: "The goat coughs at night in the field overhead"; "The woman keeps the kitchen, makes tea/Sneezes at evening, poking the peevish gutter"; "I have lost my sight, smell, hearing, taste and touch" (ll. 11, 13–14, 59). Key symptoms of the influenza are mentioned directly, including sneezing, coughing and the loss of taste and smell. Gutters were an image of abjection because they were usually filled with waste, which was considered a hotbed for infectious diseases (Barry 2005, pp. 50, 199). Eliot's personal experience of the 1918 pandemic also finds its way into the poetic details. In the *Dial* in July 1921, Eliot (1921, p. 213) described his embodied experience of the influenza as being placed in a "hot rainless spring", leaving "extreme dryness and bitter taste in the mouth"."Gerontion" depicts a very similar setting at the beginning of the poem: "Here I am, an old man in a dry month/Being read to by a boy waiting for rain" (ll. 1–2). The association between disease and Jewish people is evident in the following lines: "And the Jew squats on the window sill, the owner/Spawned in some estaminet of Antwerp/Blistered in Brussels, patched and peeled in London" (ll. 8–10). Once again, the poem establishes the sexuality-disease-Jewish conspiracy connection. The reference to Belgium has certain historical resonances. After the German occupation of Belgium during WWI, a large number of women lost their jobs and became prostitutes, and were stigmatized as the transmitters of venereal disease among

troops on both sides (Gubin et al. 2016, p. 4). The poem further conveys a misogynistic and anti-Semitic message: the Jewish landlord reproduces not only sexually but also transmits disease from Belgium to London. By highlighting that the landlord is Jewish, the poem adds the issues of money and conspiracy into its concerns.

The invasive power of money is described in the form of disease, which is regarded as spread by Jewish conspirators. In "Gerontion", Jewish landlords can invade every corner through their money power and drive people into an abject state of homelessness: "My house is a decayed house/and the Jew squats on the window sill, the owner" (ll. 7–8). The poetic persona repetitively describes his homelessness in terms of wind: "I an old man/A dull head among windy spaces"; "Vacant shutters/weave the wind. I have no ghosts/An old man in a draughty house/Under a windy knob"; "Tenants of the house/Thoughts of a dry brain in a dry season" (ll. 15–16, 29–32, 74–75). The quoted lines all highlight the abject situation in which the boundary of the speaker's house seems to be completely penetrated by the wind. The poetic speaker's identity as a tenant suggests the constant threats of homelessness. Hence, there is a parallel between the wind and the Jewish landlord's money as both can invade and occupy the spaces of the house and drive the tenant away. In the discussion of the historical background of the 1918 pandemic, the connection between winds and infectious diseases is established. Hence, it seems that Eliot shows how money conspires with the spreading of the disease through the squeezed living space of poor tenants. It would have been a brilliant critique of capitalism's ruthless exploitation during the pandemic. However, such a critique becomes extremely problematic when he simply targets a specific community as the scapegoat of capitalism. As discussed earlier, the anti-Semitic imagination of a Jewish conspiracy behind the war was gaining more and more popularity from 1918 to 1919. For example, in another poem "A cooking egg", Eliot mentions Alfred Mond who was believed to be a Jewish war-profiteer (Eliot et al. 2015, vol. 1, p. 510). In "Gerontion", Eliot describes history as a matter closely bound up with the misogynistic and anti-Semitic imagination of the "Jewish money power".

History is imagined as a deceptive woman who tried to sell her story to others:

History has many cunning passages, contrived corridors

And issues, deceives with whispering ambitions,

Guides us by vanities. Think now

She gives when our attention is distracted

And what she gives, gives with such supple confusions

That the giving famishes the craving (ll. 35–39).

History is personified as a woman who seduces and deceives others with misguiding signals. Feminine temptation is depicted as matter of give-and-take in business. In the poem there always seems to be an information gap between the woman/history and "us" (the group with which the speaker identifies). The personified history as a woman plays with people's beliefs in her words and thus there seems to be no fair or reciprocal exchange of beliefs between the woman/History and "us". The link between beliefs and give-and-take is established in the realm of finance. "History" tries to sell her stories to cater to the audiences' vanities and ambition in a confusing and cunning way. Eliot does not specifically tell the readers what kind of cunning passages and stories are discussed in this poem. However, we can infer from the use of the words "ambitions", "passion", "heroism" and "courage" that the passages are concerned with the propaganda of war (ll. 41 45, 46). Considering the financial undertone, Eliot subtly establishes a connection between people's misbeliefs in war propaganda and the unreliable financial credits that promised to give back in the future. The debt crisis of European countries after WWI broke down the illusion of a promising future. Instead, the situation of high debts only "famishes the craving". The wording here suggests the historical situation of financial breakdown of credits and its relation to famines in Europe. Eliot's familiarity with Keynes's works after his employment in Lloyds Bank facilitated his understanding of the high debts of European countries. Among them, Austria was bankrupt with severe famine in the country. Eliot

also expressed genuine concerns for the destitution in Europe, especially famine in Austria in his letter on December 18th, 1919 (Eliot et al. 2009). The poem proceeds to describe the capitalist logic of the ideal timing for investment in front of the dire situation of famine: "Gives too late/What's not believed in, or is still believed/In memory only, reconsidered passion. Gives too soon/Into weak hands/what's thought can be dispensed with/Till the refusal propagates a fear" (ll. 39–43). If the credits were given too late, the targeted receiver may have lost the passion in this investment and have gone to find others. If too early, it may not be the maximum profit for the investor. Patricia Clavin discussed similar financial calculations of other countries concerning famine and bankruptcy in Austria (Clavin 2014, pp. 270–74). Britain, France and the U.S. all calculated from their own national interests, which led to the failure in cooperation to save Austria's destitution in 1919.

In the poem, history is compared with a deceptive woman who plays with people's beliefs and exploits their feelings for her advantage. Eliot's image of history seems, disturbingly, similar to a cunning financial lender who never realizes what she promises and keeps impoverishing others. In other words, she is a war-profiteer, which corresponds to the image of the feminized Jewish landlord. The poetic persona realizes that the credits of the female history are failing, or even bankrupt. Hence, the feminization of history mirrors the rhetoric of motherland and the patriotic faith in nationalist narratives. During WWI, British propagan dare sorted to the gendered description of motherland and targeted the masculine audience for its defense of the country (Caris 2015, p. 56). The war propaganda assigns the role of defender to men who are thought to be dedicated and loyal to their family and nation. The implied message is that their nation is a feminized homeland. Britannia is the feminized homeland (motherland)in the nationalistic propaganda. Eliot is disillusioned with the ideal of masculinity and the nation's image as a stable "motherland". He does not directly question the ideal of masculinity in war propaganda. Instead, he turns the mother-figure into an untrustworthy woman and expresses his distrust of the loyal feminine image of war-time nationalism. The poetic persona mentions the father-figure, but the mother-figure is absent, and femininity is associated with seduction, deception and animalistic sexuality in "spawning": "Unnatural vices/are fathered by our heroism" (ll. 44–45). The parallel between father and heroism mirrors the imagination of masculinity during the war but "unnatural vices" reveals a sense of disillusionment with the credits of the "motherland". The disappointment in the nationalist narrative of history is expressed through the invasive power of capital in a cosmopolitan world. The poem implies that the "good mother-wife" figure has been turned into a deceptive woman by the "Jewish money power".

The poem mentions that Gerontion is "driven by the trades" to a house with multinational renters:

> And an old man driven by the Trades,
>
> To a sleepy corner (ll. 72–73).
>
> Among whispers by Mr. Silvero
>
> With caressing hands, at Limoges
>
> Who walked all night in the next room;
>
> By Hakagawa, bowing among the Titians;
>
> By Madame de Tornquist, in the dark room (ll. 23–27).

The poem structures the cosmopolitan experience through Christianity and brings in the binary between Christ and Judas Iscariot:

> In the juvescence of the year,
>
> Came Christ the tiger.
>
> In depraved May, dogwood and chestnut, flowering judas
>
> To be eaten, to be divided, to be drunk

The poem chooses names that seem to suggest different national identities. These figures all share the same rental space with the poetic persona. The poem's reference to Christ invites the readers to consider the flowering Judas as a reference to Judas Iscariot who is eaten, divided and drunk by the tenants in the same way as Christ's symbolic body (bread) and blood (wine) are divided, eaten and drunk during the Holy Communion. Eliot shows the rootless cosmopolitan tenants in an abject living space of "a decayed house" with "peevish gutter" (ll. 7, 14). The poem suggests that they worship the wrong God (Judas) and therefore are driven by money and international trade to such an abject space. The national boundaries are torn down by money. Meanwhile, Judas has been regarded as the prototype of "greedy Jewish people", as Anthony Julius presents in his book (Julius 2010, p. 264). The anti-Semitic message is again clear: Jewish money power pollutes people across different nations. The association between Jewish money power and polluting abjection is established through the act of eating and drinking. The quoted lines above reveal the confusion between Judas and Christ. Christ is envisioned as a tiger that "springs in the new year" and devours the poetic speaker (ll. 20, 48). However, the word "spring" seems to refer to the "flowering judas". Judas is mistaken for Christ and is divided, eaten and drunk by the tenants as if in the Holy Communion. The poetic persona confuses the act of eating with being eaten. The fear of being devoured by the disgusting food is related to Kristeva's understanding of abjection. The poem establishes the connection between Jewish people and the disease-spreading food because it describes the Jewish landlord's "blistering" skin and then immediately shifts to the scene of a woman sneezing and making tea in a filthy environment: "The goat coughs at night in the field overhead/Rocks, moss, stonecrop, iron, merds/The woman keeps the kitchen, makes tea/Sneezes at evening, poking the peevish gutter" (ll. 11–14). In such an abjectly infectious environment, food and drink can easily spread the illness to the tenants. The tenants' bodies and the house are all in an abject state because of the invasion of the money/disease.

As pointed out earlier, the poem describes the detailed embodied effects during the 1918 pandemic, including all the flu-like symptoms. The poem also shows how the illness occupies the tenants' bodies in a way similar to how the landlord, or broadly the unjust financial system, exploits them:

> I have lost my passion: why should I need to keep it
>
> Since what is kept must be adulterated?
>
> I have lost my sight, smell, hearing, taste and touch:
>
> How should I use them for your closer contact (ll. 57–60),
>
> These with a thousand deliberations
>
> Protract the profit of their chilled delirium,
>
> Excite the membrane, when the sense has cooled,
>
> With pungent sauces, multiply variety
>
> In a wilderness of mirrors (ll. 61–65).

The quoted lines show that illness has taken away the poetic speaker's sensory perceptions. He does not possess his own body and he believes that his possession will be "adulterated". The sexual implication here can be related to the misogynistic depiction of the feminized Jewish landlord. The loss of self-possession is a symptom of alienation in capitalism. Even in such an immobile state, landlords can "protract the profit" of the poetic persona's weak body such as the flu virus that gradually eats away his body. In this moment of the poem, the poetic speaker's immunity completely breaks down in its battle against the virus that has taken over his body, just as capitalism invades the rented space of the house. The ill-functioning immune system can also be seen in the biological and medicinal description of membranes in the poem. Membranes often refer to the outer layers of cells or bodily tissues, which are the micro-level frontiers of the immune system (Barry 2005, p. 104). The response to pungent sauces of the poetic speaker's body suggests

the failure in the functioning of the defensive immune system. The "chilled delirium" and cooled sense imply that the awaiting result is death:

> Gull against the wind, in the windy straits
>
> Of Belle Isle, or running on the Horn
>
> White feathers in the snow, the Gulf claims
>
> An old man driven by the Trades
>
> To a sleepy corner (ll. 69–73).

The poetic speaker's house gradually disappears into images of complete coldness. Since the house is a metaphor for his body, the poetic speaker seems also to gradually lose consciousness and sleeps in a corner of coldness. The poetic speaker also talks about the post-mortem image of his cold and stiffened body: "when I, stiffen in a rented house" (ll. 50). "Wilderness of mirrors" is an expression that captures the tenant's miserable state of abjection (the breakdown of the boundary between home and wilderness) and loneliness (self-mirrored images seem only to reinforce the sense of isolation) in his final moment of life (ll. 65). The wild life of spiders and weevils reveals the disintegration of the boundary of his home (ll. 65–67). The "fractured atoms" also pronounce the death of immunity and the end of the home and the body as living organisms (ll. 69).

The quoted lines above show that the poetic persona finishes his monologue in the end and the poem shifts into an objective description of the gulf. The poem also gives voice to the Gulf through the speech-act of claiming "the white feathers in the snow" (ll. 71). The voicing of the Gulf seems to say that although it is hard to discern the whiteness of feathers in the land of snow, the Gulf nonetheless knows and marks that the feathers are the only trace of an individual's life. The poetic persona's abject life in a windy and decayed house is similar to the gull against the wind. His abject life seems to be effaced easily during the pandemic just as the indiscernible white feathers in the snow.

The last two lines of the poem can also be read as an attempt to mark the short and vulnerable lives of the tenants during the pandemic: "Tenants of the house/Thoughts of a dry brain in a dry season" (ll. 74–75). These last two lines are written at the right margins of the page, which read as the authors' signature. The signature suggests that the poem is a collective creation by the suffering tenants in general during the pandemic. According to the briefing by Wendy Wilson on the short history of rent control in the U.K. during the First World War, there were rental strikes from time to time during the war (Wilson 2017, p. 4). Rent control was introduced in the U.K. in 1915 to prevent war-profiteering landlords (ibid.). The rent crisis was associated with the anti-Semitic stereotype of Jewish landlords as war-profiteers. The stereotypical reference to Jewish war-profiteering is also present in Eliot's poem "A Cooking Egg":

> I shall not want Capital in Heaven
>
> For I shall meet Sir Alfred Mond.
>
> We two shall lie together, lapt
>
> In a five per cent. Exchequer Bond (ll. 14–16).

Sir Alfred Mond was a Jewish industrialist and MP who produced munitions that accidentally exploded in East London in 1917 (Eliot et al. 2015, vol. 1, p. 510). The quoted lines above reveal Eliot's ironic criticism of war-profiteering capitalism. Eliot's "Gerontion" also expresses sympathy towards the tenants and depicts the financial housing crisis in that historical period. However, Eliot stereotypically associates Jewish identity with greed and war-profiteering, as in the degrading description of the Jewish landlord in "Gerontion". The emphasis on tenants also reinforces the anti-Semitic hostility towards an image of "the Jewish landlord". His critique of capitalism is directed to women and the Jewish community. Hence, the collective signature by the tenants at the end of the poem excludes women or the Jewish community. The misogynistic and anti-Semitic implication in such a collective creation should not be neglected.

Since Anthony Julius' systematic research on Eliot and anti-Semitism, xenophobia in Eliot's works became a heated topic in modernist studies. As the article has shown, the 1918 pandemic facilitated the spread of colonialism and anti-Semitism in the society in that period. Eliot's works reveal that he also fell into the racist and prejudicial position. Given the current COVID-19 pandemic, it is significant to point out and prevent such prejudices that could easily develop into hate speeches in the literature.

**Funding:** This research received no external funding.

**Acknowledgments:** I am grateful to my supervisor Paul Crosthwaite for providing extensive feedbacks on my original article.

**Conflicts of Interest:** The author declares no conflict of interest.

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
