# Peer review of "T.S. Eliot in the 1918 Pandemic: Abjection and Immunity"

_2410-9789, doi:10.3390/literature2020008_

Round 1
Reviewer 1 Report
This is a strong essay. The English is sometimes unidiomatic but generally clear. The thesis is a bold one, and the argument sustainable, particularly with regard to the discussion of Eliot's "Tradition and the Individual Talent." The discussion of images relating to anti-Semitism and empire is less clear when it comes to the Sweeney poem and "Gerontion," as the sequences in these poems need explication generally to allow the imagery to be contextualized. In "Sweeney," Sweeney and "The Vertebrate" are conceived as separate entities, though I would read the poem as using "vertebrate" to refer to Sweeney, who is thus classed was among those creatures defined by having a backbone. Sweeney is anti-heroic as a sordid soldier soliciting prostitutes and subject to plague and death. "Gerontion" could use framing as to its general intent. The analysis of the role of capital is good. If the author hasn't read Crawford's Young Eliot, I recommend he or she do so--the tie-in with infidelity in Eliot's household might be of interest, as would be the discussion of the Colombo poems, Eliot's work in the bank, and other concerns. I cannot speak at all to the accuracy of the representation of medical knowledge in the 1910s, as this is outside my area, so I can say nothing about their accuracy. I would recommend that the author bring in a more conclusive final paragraph or section to return us to the theme of pandemic and its tie-in with colonialism and anti-Semitism, markers at the macro level of the racist and prejudicial positions brought on by the pandemic, just as in our own day Covid aroused suspicion and hatred against Asians.
Following are some notes I took, largely stylistic, using a system of page number/line number.
¼. and 1/21.
leads academic attention
1/21==mention author, no need for “book”
1/25 mourns deaths
1/29—modernism (why plural?)
3/95 Virus stands on the biological/chemical, organic/inorganic and alive/death boundary. rephrase? The virus… OR Viruses stood on…
3/213—effective instead of affective?
9/343 Eliot’s own image and his similar. Eliot’s own image and that of similar
9.345 The poetic personas in Eliot’s poems, are also--The poetic personas in Eliot’s poems are also
9/347 means the extinction of personal life into chemical substance of a –into the chemical substance..
10/372 adapts to different ages by different. rephrase. Authors may create or define ages but are not identical to them
10/406—Eliot HAD seen
10/427 COMMENT—Sacre Cœur in Paris also is in Montmartre, often associated with brothels
14/561--The fear of “feminine abjection” _can thus be seen as the fear of empire—clarify how empire is feared—fear of having, retaining, losing, losing face….
- desire for resources, but also fear of loss of resources
14/571—seems to reproduces –seems to reproduce
17/746 as a matter
top of 18/750
come back to pandemic in the end?
Author Response
Thank you so much for your time and efforts in reviewing my essay. Your feedbacks are helpful for me to improve my research. I have made certain revisions to my essay according to your comments.
First I agree that the sequences in Eliot's poems are very vague and thus it makes the contextualization difficult. If there are places in the poem that I should contextualize more clearly, please let me know and I will try my best to explain more clearly.
In terms of "the vertebrate", I agree with your reading but I would also like to insist on my own interpretation in the essay because whether the vertebrate and Sweeney are the same person is not explicitly told in the poem.
Crawford's book is brilliantly written and yes the tie-in with the infidelity issue in Eliot's household is interesting. I would need some more time to re-visit the book and include the details in my essay.
I have already added a paragraph of conclusion according to your advice and have also corrected the stylistic inaccuracies. There is only one place ("top of 18/750") that I do not really understand what the comment asks me to do. if you could clarify about it I would really appreciate it.
Thanks again for your review.
Reviewer 2 Report
The author situates their paper within current Eliot scholarship and offers a significant, novel contribution that deftly balances the establishment of historical context and close readings of primary texts. The paper's argument is also relevant with respect to contemporary life.
Before publication, I recommend that the author make some small revisions. Primarily, I suggest removing the paragraphs on Kristeva on page 2. The passage is out of place. More importantly, by placing this passage before the thesis, the author suggests that Kristeva will play a major role in the argument and she does not. There is one minor reference on page 7 and a few sentences on pg. 14-15. The engagement with her work is not a significant part of the argument, and that is fine. But that needs to be clarified. The argument stands on its own merits without this appeal to Kristeva.
On a more minor note, I would recommend revising the phrase used in the abstract and introduction, "leads the academic attention." I understand what the author means but this is not a phrase I've ever heard used before. I would reword this.
Also, there are a number of copyediting changes that should be made. For example, pg. 1 line 29: modernisms (likely intended to be singular), pg. 1 line 40: a missing article, pg. 1 line 32: pandemics - if the author means both the 1918 and Covid-19 pandemics, that needs to be made more clear. Or, perhaps they intended it to be singular, etc.
Again, this is a strong paper.
Author Response
Thank you for your efforts in reviewing my article. I really appreciate it that you like my essay and I have revised it according to your comments.
I agree with your comment on "Kristeva" and have deleted some paragraphs accordingly on pg.2. I have also corrected the grammatical inaccuracies except one place ("pg. 1 line 40: a missing article") because I fail to see where should I insert a missing article. I would really appreciate it if you could explain more about it to me.